# Aerodynamic Effects of a Blended Multi-Winglet on an Airliner in Subsonic and Transonic Regimes

**DOI:** 10.3390/biomimetics10080522

**Published:** 2025-08-10

**Authors:** Erina Kobayashi, Kazuhisa Chiba, Wataru Yamazaki, Masahiro Kanazaki

**Affiliations:** 1Graduate School of Informatics and Engineering, The University of Electro-Communications, 1-5-1, Chofugaoka, Chofu 182-8585, Tokyo, Japan; kazchiba@uec.ac.jp; 2Department of Mechanical Engineering, Nagaoka University of Technology, 1603-1, Kami-Tomioka, Nagaoka 940-2188, Niigata, Japan; yamazaki@mech.nagaokaut.ac.jp; 3Graduate School of System Design, Tokyo Metropolitan University, 6-6, Asahigaoka, Hino 191-0065, Tokyo, Japan; kana@tmu.ac.jp

**Keywords:** multiplication of blended single-winglet, subsonic and transonic speeds, aerodynamic characteristics, wake vortex intensity, drag decomposition, computational fluid dynamics

## Abstract

This study investigates the aerodynamic performance of a blended multi-winglet configuration installed on the wingtip of a transonic commercial aircraft, focusing on both subsonic and transonic regimes. Conventional single winglets are typically optimized to reduce induced drag during cruise, but multi-winglets have the potential to enhance lift during takeoff and landing. However, their effectiveness in transonic conditions remains insufficiently explored. In this work, a reference Boeing 767 blended winglet was divided into three distinct elements, each retaining the original wingtip airfoil. Computational simulations were conducted to compare single- and multi-winglet configurations under cruise conditions. Additional analyses were performed at subsonic speeds to evaluate lift performance. Under transonic conditions, the multi-winglet configuration exhibited a 1.4% increase in total drag due to a greater projected frontal area. However, it achieved reduced induced drag, attributed to the rearmost winglet’s negative cant angle, which suppresses vortex formation by inhibiting upward airflow. In subsonic flight, lift improved by up to 1.3% due to accelerated flow over the upper surface, enhanced by smaller leading-edge radii and air acceleration through inter-winglet gaps. These findings suggest that multi-winglets outperform single winglets in reducing induced drag during cruise and enhancing lift during takeoff and landing.

## 1. Introduction

The growing urgency to address climate change has underscored the need for environmentally sustainable aircraft design. Among various strategies, reducing aerodynamic drag remains one of the most effective means of improving fuel efficiency. The mid-field drag decomposition method classifies total drag into wave, profile, and induced components [1], with induced drag accounting for approximately 40% of the total drag during cruise and up to 80–90% during takeoff [2,3]. Winglets, which mitigate wingtip vortices, offer a practical and widely adopted solution to reduce induced drag, thereby enhancing overall aerodynamic efficiency. Consequently, winglet geometry optimization continues to be a focal point in aerodynamic research.

Whitcomb’s pioneering wind tunnel tests on a narrow-body jet equipped with winglets demonstrated a 20% reduction in induced drag and a 9% increase in the lift-to-drag ratio L/D [4]. These findings prompted extensive investigations into winglet design, focusing on geometric parameters such as dihedral, twist, and sweep angles [5,6]. Elham et al. [7] applied multi-objective optimization to winglets, considering drag and structural weight, achieving a 3.8% reduction in fuel weight for the Boeing 747. Further developments include morphing winglets that actively adapt their cant and twist angles to optimize performance across flight conditions [8,9,10,11].

Split-tip winglets with auxiliary surfaces mounted on the lower side of the main winglet have also been explored [12,13,14]. Reddy et al. [12] conducted multi-objective optimization of such configurations, achieving a 2.7% reduction in the drag coefficient $C_{D}$ compared to conventional single winglets.

Biomimetic winglet concepts, inspired by bird flight, have also received considerable attention [15,16]. Birds reduce drag using independently movable primary feathers to disrupt tip vortices [17]. Analogously, multi-winglet configurations comprising multiple chordwise-mounted elements have been proposed. Smith et al. [15] demonstrated in low-speed wind tunnel tests that five planar winglets produced discrete vortices, reduced drag, and improved L/D by up to 30% relative to a wing without winglets. Shelton et al. [18] found that actively adjusting the angle of attack α and dihedral angle of each winglet could replace ailerons and improve gust load alleviation. Miklosovic et al. [19] showed, through vortex lattice methods and experiments, that modifying cant angle combinations could reduce $C_{D}$ by 54% and increase L/D by 57%, while also enhancing longitudinal stability. Additional studies have investigated the aerodynamic benefits of twist and sweep [20], as well as increasing the number of winglets [21,22].

Beyond drag reduction, multi-winglets also show promise in enhancing lift. Lynch et al. [23] investigated spacing between winglets as an analog to gaps between raptor feathers. Wind tunnel testing of a three-winglet configuration at 10.1 [m/s] demonstrated average and peak lift coefficient $C_{L}$ increases of 7.3% and 5.6%, respectively. Reddy et al. [24] optimized a multi-winglet configuration for a commercial aircraft under takeoff and landing conditions (Mach number *M* of 0.25, $\alpha =11^{\circ }$), achieving a 4% reduction in $C_{D}$ and an 8% increase in $C_{L}$ compared to a blended winglet. These findings indicate that multi-winglets can simultaneously enhance lift and reduce drag, while also offering the potential to mitigate wake turbulence through the redistribution and diffusion of wingtip vortices.

Nonetheless, most prior research has focused on subsonic applications. Their effectiveness in transonic regimes and on full-scale commercial aircraft remains inadequately understood, and the associated flow physics are not yet fully clarified. This is particularly important due to differing flow physics across regimes, such as shock-induced separation in transonic conditions and vortex roll-up under subsonic operation. Demonstrating the aerodynamic effects of the multi-winglet under subsonic and transonic conditions indicates that avian-inspired characteristics can broaden the applicability of winglets. Furthermore, it suggests that biomimetic technologies can be extended to artificial structures and operational environments far exceeding the size and natural habitat of the organisms being mimicked. Therefore, these findings are expected to provide valuable insights not only for advancing aircraft design but also for biomimetics.

This study investigates the $C_{D}$ reduction effects of multi-winglets under transonic conditions, aiming to support their application in the cruise phase of a transonic aircraft. To complement this, analyses are also conducted under subsonic conditions representative of takeoff and landing. These analyses seek to clarify the previously unexplained mechanisms behind $C_{L}$ enhancement. These insights can provide design guidelines for further $C_{L}$ improvement during takeoff and landing, and are applicable not only to transonic aircraft but also to subsonic configurations. Accordingly, we examine the aerodynamic performance of a blended multi-winglet installed on the NASA Common Research Model (CRM) [25] under both transonic and subsonic flight conditions. Two CRM configurations are analyzed: one with a conventional blended single winglet and another with a blended multi-winglet comprising three elements.

The remainder of the paper is structured as follows. Section 2 describes the computational setup, including geometry, mesh generation, and numerical methodology. Section 3 and Section 4 present and analyze the results: Section 3 focuses on transonic drag decomposition, while Section 4 discusses lift enhancement and wake turbulence mitigation under subsonic conditions. Section 5 summarizes the main conclusions and outlines future research directions.

## 2. Problem Setup

### 2.1. Configurations

When the aerodynamic characteristics of the wing and fuselage are well established, the aerodynamic contribution of winglets can be isolated. Accordingly, this study employs the sixth Drag Prediction Workshop (DPW6) configuration of the NASA CRM, which comprises only the wing and fuselage [26]. All winglet geometries were designed using Autodesk Fusion 360 (“Autodesk Fusion” available online at https://www.autodesk.co.jp/products/fusion-360/ [accessed: 7 November 2024]). Detailed geometric parameters are provided below.

#### 2.1.1. Single-Winglet Configuration

The reference winglet geometry is based on the blended winglet of the Boeing 767-300 (B767), selected for its dimensional similarity to the NASA CRM. Among commercial aircraft with publicly available specifications, the B767 has a half-span length most comparable to that of the NASA CRM, making its winglet geometry particularly suitable.

However, since the tip chord length and span of the main wing differ between the CRM and the B767, the dimensions were defined such that the height-to-span ratio h/S and the taper ratio $\lambda =c_{\mathrm{tip}}/c_{\mathrm{root}}$ match those of the B767, as shown in Figure 1. The cant angle θ and sweep angle ψ were retained from the original design. This approach ensures geometric consistency with the original B767 winglet while adapting to the CRM’s wing planform. This modified geometry is hereafter referred to as the “single winglet.” Its dimensions are listed in Table 1.

#### 2.1.2. Multi-Winglet Configuration

As illustrated in Figure 2, the “multi-winglet” configuration was derived by dividing the single winglet into three chordwise segments, mimicking the morphology of bird primary feathers. This represents the minimum number of winglets required to capture the aerodynamic influences from both the forward and rearward winglets. While additional elements may further improve performance, the present study aims to isolate and understand the minimum configuration that exhibits both forward and rearward aerodynamic effects. Each segment is smoothly connected to the main wing to mitigate increases in interference drag. For brevity and clarity, the winglets are hereafter referred to as the first, second, and third winglets in the order from the leading edge to the trailing edge. The geometric design is summarized below.

##### Airfoil

Each segment’s cross-sectional shape is identical to that of the main wingtip. The root chord $c_{\mathrm{root}}$ of each segment equals the chord length of the wingtip, and the chord length of each segment is one-third that of the single winglet.

##### Cant Angle

As shown in Figure 2a, the cant angles of the first, second, and third segments are set to $\theta_{1}=45^{\circ }$, $\theta_{2}=15^{\circ }$, and $\theta_{3}=-15^{\circ }$, respectively. These values are adopted from the wind tunnel experiments by Catalano et al. [27], where this combination yielded the highest subsonic $C_{L}$ among 54 tested configuration. However, it should be noted that this arrangement is not necessarily optimal for the present study due to significant geometric differences. The objective here is to assess the aerodynamic advantages of the multi-winglet relative to the single winglet under both subsonic and transonic conditions and to elucidate the associated flow mechanisms; detailed design optimization is left for future investigation.

### 2.2. Mesh Generation

Unstructured meshes were generated using MEGG3D ver.6.9.0.7 [28,29,30,31,32,33,34,35,36,37,38], which enables automated mesh generation with high flexibility. Prismatic layers were applied near the surface to resolve boundary layer flows accurately. The first-layer thickness, stretching factor, and the maximum number of prism layers were set to 0.01/Re, 1.2, and 45, respectively. Re signifies Reynolds number. The computational domain was defined as a hemisphere with a radius 15 times the fuselage length.

To evaluate the impact of wake turbulence and induced drag via the drag decomposition method, mesh refinement was applied in the wake region near the wingtips to capture the wingtip vortices accurately. The refined regions are depicted in Figure 3. In addition to the region enclosed by the dotted line, a broader area was subdivided into three progressive levels to prevent abrupt transitions in mesh size. In the streamwise direction (*x*), the domain extended 200 [m] downstream from the main wing’s leading edge (approximately 3.2 times the fuselage length *L*). Although full wake development would require far-field refinement, mesh resolution was limited to the near-field to control computational cost, given that wake mitigation is a secondary objective of the multi-winglet. In the lateral (*y*) and vertical (*z*) directions, the refined region spans a 35 [m] square centered at the single-winglet tip, ensuring the entire dispersed vortex is captured downstream.

The mesh for the single-winglet model comprised approximately 0.61 million surface nodes and 40.8 million volume nodes. The multi-winglet model contained 0.73 million surface nodes and 43.6 million volume nodes.

### 2.3. Flow Solver

Simulations were conducted using FaSTAR ver.6.0.7 [39], a compressible flow solver developed by the Japan Aerospace Exploration Agency. The solver employs an unstructured node-centered finite volume method with MUSCL-type reconstruction [40] to solve the following three-dimensional Reynolds-averaged Navier–Stokes (RANS) equations described in integral form:(1)$$\frac{\partial }{\partial t}\int_{\Omega }QdV+\int_{\partial \Omega }\left\{F\left(Q\right)-G\left(Q\right)\right\}\cdot ndS=0,$$
where(2)$$Q={\left[\begin{array}{ccccc} \rho & \rho u & \rho v & \rho w & e \end{array}\right]}^{⊤}$$
is the vector of conservative variables. ρ denotes the density, *u*, *v*, and *w* are the velocity components in the *x*, *y*, and *z* directions, respectively, and *e* represents the total energy. F(Q) and G(Q) are the inviscid and viscous flux vectors, respectively. n is the unit normal vector on the boundary ∂Ω of the control volume Ω. Second-order spatial accuracy is ensured by applying Hishida’s differentiable slope limiter [41]. The Harten-Lax-van Leer-Einfeldt-Wada (HLLEW) scheme [42] was used for numerical flux calculation, and the Lower–Upper Symmetric Gauss–Seidel (LUSGS) scheme [43] was employed for time integration. The Explicit Algebraic Reynolds Stress Model with *k*-ω formulation (EARSM) [44,45] was used for subsonic conditions, while the Shear Stress Transport (SST) *k*-ω turbulence model [46] was selected for transonic cases to better capture boundary layer transitions. This study focuses on qualitatively elucidating the aerodynamic effects of the multi-winglet under each condition. Therefore, the present analysis is limited to validation using these turbulence models. Comparative verification using multiple turbulence models will be conducted in future work for quantitative evaluation.

All computations were performed on a local cluster comprising five nodes, each equipped with an Intel Xeon E5-2660 (2.2 GHz, 16 cores) and 64 GB RAM. The computational time for the subsonic simulations was approximately 106 h for the single-winglet configuration and 116 h for the multi-winglet configuration. For transonic simulations, the runtimes were 91 and 95 h, respectively.

### 2.4. Computational Conditions

The boundary conditions were defined as follows. A uniform freestream velocity was imposed at the far-field boundaries, corresponding to the specified Mach number and angle of attack for each test case. Turbulence quantities for the *k*-ω models were set to default freestream values (k=0.001, ω=1.0) unless otherwise noted. All solid surfaces, including the wing and fuselage, were treated with no-slip and adiabatic wall boundary conditions. The outlet boundaries were extrapolated with zero-gradient conditions. Wall functions were not used; the first cell height was set to resolve the viscous sublayer directly (y+<1). The simulations assumed transonic conditions for cruise and subsonic conditions representative of takeoff and landing. Subsonic conditions were based on the fourth American Institute of Aeronautics and Astronautics (AIAA) Computational Fluid Dynamics (CFD) High-Lift Prediction Workshop (HLPW4) [47], while transonic conditions followed the setup of the sixth AIAA CFD Drag Prediction Workshop (DPW6) [26]. *M*, α, and Re for each case are summarized in Table 2. Since the objective of this study is to clarify the aerodynamic effects of the multi-winglet within the broad speed regimes of transonic and subsonic conditions, the analysis is limited to two *M* commonly employed in aerodynamic studies of the CRM.

## 3. Numerical Results: Transonic Aerodynamics

A principal function of winglets is to reduce induced drag during transonic cruise. This section evaluates the effectiveness of a multi-winglet configuration in achieving this objective, employing a drag decomposition approach. The results indicate that the multi-winglet configuration reduces induced drag relative to the single-winglet baseline. However, increases in profile and wave drag lead to a net rise in $C_{D}$ of approximately three counts at the cruise α. Nevertheless, since $C_{L}$ also increases concurrently, the overall L/D remains comparable to that of the single-winglet configuration.

### 3.1. Aerodynamic Discrepancies

Figure 4 presents the aerodynamic performance under transonic conditions, along with the differences in aerodynamic coefficients between the single- and multi-winglet configurations, defined as follows:(3a)$$\begin{array}{cc} \Delta C_{L} & ≜{C_{L}}^{\left(\mathrm{multi}\right)}-{C_{L}}^{\left(\mathrm{single}\right)}, \end{array}$$(3b)$$\begin{array}{cc} \Delta C_{D} & ≜{C_{D}}^{\left(\mathrm{multi}\right)}-{C_{D}}^{\left(\mathrm{single}\right)}, \end{array}$$(3c)$$\begin{array}{cc} \Delta (L/D) & ≜{\left(L/D\right)}^{\left(\mathrm{multi}\right)}-{\left(L/D\right)}^{\left(\mathrm{single}\right)}. \end{array}$$

Figure 4a shows that the multi-winglet configuration yields a higher $C_{L}$ than the single winglet for most α. For both configurations, $C_{L}$ increases again beyond the stall angle ($\alpha =8^{\circ }$) due to a weakened shock wave near the wingtip compared to $\alpha \leq 8^{\circ }$. The reduced shock strength suppresses pressure recovery behind the shock, decreasing the pressure near the main wing’s trailing edge. This local shock weakening reduces the adverse pressure gradient along the chordwise direction, thereby delaying or partially suppressing flow separation near the wingtip. As a result, lift recovers beyond $\alpha =8^{\circ }$, particularly in the outboard sections where shock effects dominate.

As shown in Figure 4b, the $C_{D}$ is consistently higher for the multi-winglet configuration across all α. At α=2.75°, the increase amounts to 3.4 counts (1.4%). Furthermore, at the design $C_{L}=0.50$ for the NASA CRM [26], Figure 4c reveals a drag penalty of 1.8 counts (0.7%) for the multi-winglet relative to the single winglet.

Figure 4d illustrates that the difference in L/D at α=2.75° is negligible, with the multi-winglet configuration achieving a slight improvement of 0.07%. This marginal gain is attributed to the 1.3% increase in $C_{L}$, which compensates for the rise in $C_{D}$. However, additional reduction in $C_{D}$ is necessary, as the higher lift near the wingtip also induces a greater rolling moment. In the subsequent section, each drag component is individually examined to identify the sources contributing to the increase in $C_{D}$.

### 3.2. Drag Decomposition Results

This subsection compares the two configurations by categorizing the constituent components of drag. The objectives are to identify the factors contributing to the increase in $C_{D}$ for the multi-winglet configuration and to assess its effectiveness in reducing the induced drag coefficient $C_{D_{i}}$ under cruise conditions. To this end, the drag decomposition method is applied at the cruising α, partitioning the total drag into profile drag ($C_{D_{pr}}$), wave drag ($C_{D_{w}}$), induced drag ($C_{D_{i}}$), and spurious drag ($C_{D_{s}}$). The profile drag includes contributions from boundary layers on solid surfaces and the aircraft wake. The wave drag accounts for pressure drag associated with shock waves, and the induced drag results from lift generation. The spurious drag term represents non-physical contributions, often arising from mesh-induced errors or numerical artifacts. The mid-field drag decomposition method [1] used in this study divides the flow field into distinct regions associated with physical phenomena that contribute to drag generation. The drag components are then quantified by integrating relevant flow quantities within each region. While both $C_{D_{pr}}$ and $C_{D_{w}}$ increase for the multi-winglet, the reduction in $C_{D_{i}}$ compensates for these increases.

Figure 5 summarizes the drag decomposition results. The error bars on $C_{D_{i}}$ account for the fluctuations observed beyond x≥210[m] in Figure 6, unlike the other components ($C_{D_{pr}}$, $C_{D_{w}}$, and $C_{D_{s}}$). These fluctuations arise due to the limited resolution beyond the mesh refinement boundary at x=200[m] (Figure 3b), which impairs the accurate capture of vortex dynamics. Therefore, $C_{D_{i}}$ is spatially averaged over the range x=48[m] to x=140[m], with the maximum and minimum values indicated as error bars.

Profile Drag ($C_{D_{pr}}$): As shown in Figure 5, $C_{D_{pr}}$ increases by 2.7 counts in the multi-winglet configuration, primarily due to its larger frontal projected area.Wave Drag ($C_{D_{w}}$): $C_{D_{w}}$ also increases, by approximately 1.0 count. However, its contribution to the overall increase in $C_{D}$ is smaller than that of $C_{D_{pr}}$, owing to the relatively weak shock waves generated by the winglets.Induced Drag ($C_{D_{i}}$): $C_{D_{i}}$ is reduced by 3.0 counts in the multi-winglet configuration, and this reduction is sufficient to offset the rise in $C_{D_{pr}}$.

$C_{D_{pr}}$ and $C_{D_{i}}$ account for approximately 49% and 34% of the $C_{D}$, respectively. Hence, minimizing these components is crucial for improving aerodynamic efficiency. While an increase in $C_{D_{pr}}$ is inevitable due to the larger projected area of the multi-winglet, the reduction in $C_{D_{i}}$ suggests that further aerodynamic benefits could be realized through geometric optimization.

The next section examines vortex structures near the wingtips to clarify the mechanisms by which the multi-winglet configuration achieves a greater reduction in $C_{D_{i}}$ compared to the single winglet. These insights are expected to guide the development of optimal multi-winglet designs aimed at minimizing $C_{D}$.

### 3.3. Comparison of Vortex Structures

This subsection investigates the mechanisms by which the multi-winglet configuration reduces $C_{D_{i}}$ through a comparative analysis of vortex structures near each winglet. The objective is to derive potential design strategies for further minimizing induced drag.

Figure 7 shows isosurfaces of the second invariant of the velocity gradient tensor (the *Q*-criterion [48]) at a threshold of Q=1.0, colored by the streamwise vorticity component to indicate the direction of vortex rotation. Overall, the multi-winglet configuration attenuates the primary vortex observed in the single-winglet case and introduces several distinct vortical structures. The corresponding vortex behaviors are summarized as follows:Vortex suppression effects of the multi-winglet:The streamwise length of the wingtip vortices is shorter in the multi-winglet configuration than in the single winglet (V1 in Figure 7b).Vortex generation at the main wing-winglet junction, evident in the single-winglet configuration, is suppressed in the multi-winglet case (V2 in Figure 7b).

(See Section 3.3.1 for details.)

Vortex characteristics unique to the multi-winglet:

A vortex originates near the 70% spanwise location of the first winglet (V3 in Figure 7b).Vortices are generated along the trailing edge of the first winglet between 30% and 60% of the span, and along the second winglet between 35% and 45% of the span (V4 in Figure 7b).The streamwise extent of wingtip vortices varies across the three winglets under uniform flow conditions.

(Discussed in Section 3.3.2.)

**Figure 7 biomimetics-10-00522-f007:**
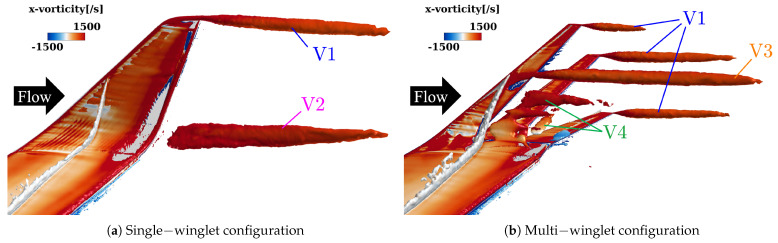
Isosurfaces of the Q—criterion at Q=1.0 with streamwise vorticity η contours at α=2.75° under transonic conditions. Warm colors denote counterclockwise−rotating vortices, while cold colors indicate clockwise−rotating vortices, as viewed from downstream. V1 to V5 are designations used to identify the vortices referenced in the text.

#### 3.3.1. Vortex Suppression Mechanism

##### Wingtip Vortex (V1)

Due to the shorter chord lengths of the individual winglets in the multi-winglet configuration, the resulting wingtip vortices are less developed and dissipate more rapidly than those in the single-winglet case.

##### Vortices at the Main Wing–Winglet Junction (V2)

As illustrated in Figure 8, spatial streamlines colored by streamwise vorticity reveal an upward flow along the outer surface of the single winglet, originating near the 50% chord position on the lower surface of the junction. In contrast, this upward flow is suppressed in the multi-winglet configuration, resulting in reduced downstream vorticity. This suppression is primarily due to the third winglet, installed at a negative incidence angle ($\theta <0^{\circ }$), which inhibits the upward flow caused by the pressure differential between the upper and lower surfaces of the main wing.

Consequently, the strength of the vortex formed at the trailing edge—where the upper and lower surface flows converge—is weakened relative to the single-winglet configuration. Thus, rear-mounted winglets are effective in mitigating vortex roll-up from the underside. To suppress vortex formation at the junction, the incidence angle of the rear winglets should be set to $\theta ≲0^{\circ }$.

#### 3.3.2. Vortex Behavior Specific to the Multi-Winglet Configuration

##### Vortex Formation near the 70% Spanwise Location of the First Winglet (V3)

Figure 9 presents *y*–*z* cross-sectional velocity vectors upstream of the vortex generated by the first winglet, overlaid with isosurfaces of the shock detection function $f_{\mathrm{shock}}=1.0$, highlighting the spatial correlation between vortex initiation and the shock wave termination. The shock function is defined as(4)$$f_{\mathrm{shock}}=\frac{V\cdot \nabla P}{a|\nabla P|},$$
where V is the velocity vector, *a* is the speed of sound, and *P* is the static pressure.

Spanwise surface flow is observed in the multi-winglet configuration but not in the single-winglet case. This spanwise flow interacts with the external flow, leading to vortex formation and is attributed to the stronger shock waves generated by the multi-winglet configuration.

In the multi-winglet model, boundary layer separation downstream of the shock promotes spanwise energy transfer driven by a lateral pressure gradient—higher near the trailing edge and lower near the leading edge. Therefore, reducing shock strength through optimized sweep and twist angles is expected to mitigate this vortex formation.

##### Vortices at the Trailing Edge of the First and Second Winglets (V4)

These vortices arise from pressure differences between the upper and lower surfaces of the main wing. As a result, the airflow beneath the first and second winglets is drawn outward—from the wing root toward the tip—as they move downstream. This effect is especially pronounced in the first winglet due to the angular relationship $\theta_{1}>\theta_{2}>\theta_{3}$, which suppresses upwash. To mitigate the formation of these trailing-edge vortices, it is therefore necessary to reduce the θ.

##### Chordwise Variation in Wingtip Vortex Extent

Among the three winglets, the first generates the shortest streamwise vortex due to its largest sweep angle ($\psi_{1}=44^{\circ }$) and correspondingly low local $C_{L}$. Interestingly, the second winglet generates the largest vortex, despite having a greater sweep angle than the third.

To explain this, Figure 10 displays velocity vectors in the *y*–*z* plane at the 50% chord location near the tips of the second and third winglets, with $v_{\mathrm{vel}}$ contours. Spanwise flow is observed on the second winglet surface but not on the third, due to the stronger spanwise pressure gradient created by the second winglet’s larger trailing-edge sweep. This intensified interaction enhances vortex formation at the second winglet tip. These findings suggest that increasing leading-edge sweep while reducing trailing-edge sweep may suppress spanwise flow and vortex development more effectively than the current design.

## 4. Numerical Results: Subsonic Aerodynamics

The multi-winglet concept was originally developed to enhance lift generation under subsonic conditions [23]. This section investigates the mechanism by which the multi-winglet configuration increases lift, based on a comparative aerodynamic analysis with a single-winglet configuration. The results show that the multi-winglet consistently achieves higher $C_{L}$ across nearly the range of α, primarily because each winglet functions as an individual airfoil, accelerating the flow over its upper surface. Additionally, an evaluation of the nondimensional circulation [49] downstream of the aircraft reveals that the multi-winglet suppresses wingtip vortex strength. These findings suggest that the multi-winglet has potential for mitigating wake turbulence, which may contribute to reducing arrival and departure separation requirements at airports.

### 4.1. Aerodynamic Discrepancies

Figure 11 presents the aerodynamic performance under subsonic conditions. Given the relevance to takeoff and landing, the discussion primarily focuses on $C_{L}$; however, the corresponding $C_{D}$ and L/D values are also shown for reference.

As depicted in Figure 11, the multi-winglet configuration enhances $C_{L}$ for $\alpha \geq 2^{\circ }$, as originally intended. Both configurations exhibit stall behavior at $\alpha =11^{{}^{\circ}}$. Prior to stall, the most significant improvement is observed at $\alpha =6^{{}^{\circ}}$, reaching an increase of 1.3%. Beyond stall, the maximum increase reaches 1.7%. Although the post-stall decline in $C_{L}$ is more gradual in the multi-winglet configuration, the overall trend remains qualitatively similar. Further investigation is necessary to determine whether an optimized multi-winglet geometry can delay stall onset or alleviate the decline in post-stall lift.

The next section examines the flow structures responsible for the observed $C_{L}$ enhancement. These insights are expected to inform design strategies for maximizing lift and delaying stall under subsonic operating conditions.

### 4.2. Mechanisms of $C_{L}$ Enhancement

This subsection elucidates the physical mechanisms responsible for the enhancement of $C_{L}$ in the multi-winglet configuration. The insights obtained serve as a foundation for developing design guidelines to further improve $C_{L}$ under subsonic conditions. The results indicate that the mechanisms contributing to $C_{L}$ enhancement differ before and after stall. The analysis first examines the flow structures at $\alpha =6^{{}^{\circ}}$, where the disparity in the $C_{L}$ growth rate between the two configurations is most pronounced prior to stall. As a supplementary analysis, the evolution and breakdown of post-stall flow are investigated at $\alpha =12^{{}^{\circ}}$ as a representative case.

#### 4.2.1. Effect of the Multi-Winglet on $C_{L}$ Enhancement Prior to Stall

Figure 12 presents the distribution of the surface pressure coefficient $C_{p}$ on the upper and lower surfaces at $\alpha =6^{{}^{\circ}}$. These results support the aerodynamic concept that each winglet functions as an independent airfoil, effectively lowering the upper surface $C_{p}$. On the lower surfaces, negative pressure regions are also observed in the first and second winglets. The following discussion separately addresses the characteristics of the upper and lower surfaces.

##### Upper Surface

As shown in Figure 12a, the upper surface $C_{p}$ of the single-winglet configuration gradually recovers from the leading edge to the trailing edge. In contrast, Figure 12b shows that the multi-winglet configuration produces localized regions of low pressure near the leading edges of each winglet. This is attributed to their distinct airfoil shapes, reduced chord lengths, and smaller leading-edge radii of curvature, all of which promote flow acceleration over the upper surfaces—particularly on the first and second winglets.

Moreover, the extent of these low-pressure regions decreases progressively from the first to the third winglet, especially near the winglet roots. To investigate the underlying cause, Figure 13 depicts streamlines colored by the vertical velocity component $w_{\mathrm{vel}}$ in the vicinity of the main wingtip. As illustrated in Figure 13a, the cant angle of the first winglet ($\theta_{1\mathrm{st}}=45^{{}^{\circ}}$) induces upward flow along its lower surface, resulting in positive $w_{\mathrm{vel}}$ ahead of the first winglet, as confirmed by Figure 13b. Given that$$\theta_{1\mathrm{st}}>\theta_{2\mathrm{nd}}=15^{{}^{\circ}}>\theta_{3\mathrm{rd}}=-15^{{}^{\circ}},$$
the magnitude of $w_{\mathrm{vel}}$ decreases from the first to the third winglet, thereby reducing the local α and weakening flow acceleration over the upper surfaces of the downstream winglets. Increasing the cant angles of the rear winglets may help recover the local α and further improve $C_{L}$. Additionally, since the local α likely varies in the spanwise direction, optimizing the twist distribution among the winglet segments may contribute to further aerodynamic enhancement.

##### Lower Surface

Figure 12c,d show that the $C_{p}$ near the trailing edges of the lower surfaces of the first and second winglets is lower in the multi-winglet configuration compared to the single winglet. This is attributed to flow acceleration over the upper surfaces of the adjacent downstream winglets (i.e., the second and third), which enhances suction effects on the lower surfaces of the upstream winglets. Although $C_{p}$ reductions are observed on both upper and lower surfaces, the more pronounced reduction on the upper surfaces results in an overall increase in $C_{L}$.

#### 4.2.2. Effect of the Multi-Winglet on $C_{L}$ Enhancement in the Post-Stall Regime

Figure 14 presents the distributions of the surface $C_{p}$ on the upper and lower surfaces at $\alpha =12^{{}^{\circ}}$. These results indicate that the formation of leading-edge vortices on the upper surfaces of the second and third winglets contributes significantly to $C_{L}$ enhancement. In contrast, the appearance of negative pressure regions near the trailing edges of the first and second winglets on the lower surface inhibits further increase in $C_{L}$. The following discussion addresses the upper and lower surface behaviors separately.

##### Upper Surface

As shown in Figure 14a, the single-winglet configuration exhibits a gradual recovery of $C_{p}$ from the leading edge to the trailing edge. In contrast, the multi-winglet configuration shows generally lower upper surface pressures, particularly in the leading-edge root regions of the second and third winglets.

Figure 15 visualizes *Q*-criterion isosurfaces and contours of the streamwise velocity component $u_{\mathrm{vel}}$ around the multi-winglet, clarifying the mechanisms responsible for these low-$C_{p}$ regions. Vortices are observed near the leading edges of the second and third winglets, resulting from partial flow reattachment. In contrast, full flow separation occurs on the first winglet. The suppression of separation on the second and third winglets is attributed to flow acceleration through the inter-winglet gaps. As the spanwise gaps widen toward the wingtip, the acceleration effect diminishes, delaying reattachment and enlarging the vortex-dominated regions. Beyond a critical gap width, acceleration ceases and the flow is deflected away from the winglet surfaces. Therefore, reducing the spanwise gap between winglets is expected to expand the flow acceleration regions and enhance $C_{L}$ on the second and third winglets.

##### Lower Surface

A comparison of Figure 14c,d shows that the $C_{p}$ near the trailing edges of the first and second winglets is lower in the multi-winglet configuration than in the single-winglet case. This drop is attributed to flow acceleration induced by the inter-winglet gaps. The effect is most pronounced on the first winglet due to the wider spacing between the first and second winglets, as illustrated in Figure 15. The underside of the third winglet is largely unaffected by this mechanism, resulting in relatively higher $C_{p}$. Consequently, the contribution of each winglet to the total $C_{L}$ increases in the order: first < second < third. Based on this trend, further improvement in $C_{L}$ may be achieved by sequentially reducing the chord lengths of the winglets in this order.

### 4.3. Effects on Wake Turbulence

This subsection investigates the effectiveness of the multi-winglet configuration in mitigating wake turbulence. During takeoff and landing, a minimum separation time must be maintained between successive aircraft to prevent hazardous interactions with the wake vortices generated by preceding aircraft [49]. With increasing air traffic demand, reducing this separation interval is becoming increasingly important. The multi-winglet design has the potential to diffuse wingtip vortices more effectively [15,22,27], thereby attenuating wake turbulence and enabling shorter arrival and departure intervals. The present results show that the multi-winglet suppresses wake turbulence more effectively than the single-winglet configuration, particularly at higher α.

Due to the inherently unsteady nature of wake turbulence, accurate prediction requires time-resolved simulations such as Large-Eddy Simulation (LES) [50]. However, because the current multi-winglet geometry remains in an early design stage, its shape has not yet been optimized, and the computational cost of high-fidelity unsteady simulations is prohibitive. In this study, time-averaged results from steady-state RANS simulations are therefore used to assess the qualitative potential of the multi-winglet for wake turbulence mitigation. Although the RANS turbulence models yield modeled turbulence quantities (e.g., turbulent kinetic energy), their magnitudes in the highly sheared, rolled-up vortex cores are strongly model- and mesh-dependent and do not provide a robust basis for comparing configurations. We therefore adopt a vorticity-based nondimensional circulation metric (Equation (Equation 5)) as a proxy for wake vortex strength, consistent with prior wake-separation assessments [51,52,53]. High-fidelity LES/DES will be pursued in future work to quantify turbulence statistics directly.

#### 4.3.1. Disparity in Wingtip Wake Characteristics Based on Nondimensional Circulation

Rapid dissipation of vortex intensity, quantified by the nondimensional circulation(5)$$\Gamma_{0}=\frac{c}{Su_{\infty }}{\int \int }_{\Sigma }\eta dydz$$
is essential for mitigating wake turbulence [49]. Here, Σ denotes the evaluation plane in the *y*–*z* plane, and the vorticity η is defined as(6)$$\eta =\frac{\partial w_{\mathrm{vel}}}{\partial y}-\frac{\partial v_{\mathrm{vel}}}{\partial z}.$$
Low-intensity, counter-rotating secondary vortices tend to form around the primary wingtip vortices, weakening their net circulation when integrated according to Equation (Equation 5). In this study, $\Gamma_{0}$ is evaluated over regions where η≥1.0, under the assumption that high-vorticity regions are more likely to develop into hazardous wake turbulence. Figure 16 shows the spatial distributions of $\Gamma_{0}$ at several downstream positions for both configurations.

Aircraft do not maintain a fixed α during takeoff and landing, and wake turbulence evolves temporally. Thus, steady-state simulations at constant α cannot fully capture its transient nature. Nevertheless, assuming that time-averaged η serves as a qualitative proxy for wake persistence and intensity, comparative evaluations are performed at four pre-stall angles of attack: $\alpha =4^{{}^{\circ}}$, $6^{{}^{\circ}}$, $8^{{}^{\circ}}$, and $10^{{}^{\circ}}$.

As shown in Figure 16, the near-field region (x≤50[m]) exhibits consistently higher $\Gamma_{0}$ values for the multi-winglet configuration across all α, due to the increased number and spatial extent of wingtip vortices. In contrast, in the far-field region (x≥55[m]), $\Gamma_{0}$ values become lower than those for the single-winglet configuration. The maximum reduction rates of $\Gamma_{0}$ in the far field are 3.1%, 4.3%, 5.2%, and 4.2% at $\alpha =4^{{}^{\circ}}$, $6^{{}^{\circ}}$, $8^{{}^{\circ}}$, and $10^{{}^{\circ}}$, respectively. Moreover, the decay rate of $\Gamma_{0}$ increases with α, indicating that the multi-winglet more effectively attenuates wingtip vortices at higher angles of attack.

#### 4.3.2. Analysis of Wake Turbulence Suppression by the Multi-Winglet Configuration

This subsection compares the vortex structures near the wingtip in both configurations to elucidate the mechanisms by which the multi-winglet mitigates wake turbulence. To minimize the influence of boundary layer transition and flow separation, the analysis is performed at the lowest α considered in this study, $\alpha =4^{{}^{\circ}}$, and the results are compared with those of the single-winglet configuration. The findings indicate that the multi-winglet suppresses not only the primary wingtip vortices but also those generated at the root of the winglet.

Figure 17 presents isosurfaces corresponding to a *Q*-criterion value of 0.05. In the single-winglet configuration (Figure 17a), prominent wingtip vortices are observed, along with secondary vortices generated at the junction between the main wing and the winglet. In contrast, in the multi-winglet configuration (Figure 17b), the wingtip vortices dissipate more rapidly, and the junction vortices are substantially weakened. These trends are consistent with those observed under transonic conditions, and the underlying suppression mechanism is considered analogous to that described in Section 3.3.1.

As shown in Figure 17b, the streamwise extent of the wingtip vortices increases sequentially from the first to the third winglet. This ordering correlates with the magnitude of each winglet’s leading-edge sweep angle; larger sweep angles correspond to lower local $C_{L}$, thereby accelerating vortex dissipation. Therefore, increasing the ψ may further promote wake vortex attenuation. Nevertheless, adequate lift generation must be maintained under takeoff and landing conditions. As previously discussed, extending the multi-winglet coverage toward the wing root—thereby increasing the spanwise portion of the main wing influenced by the multi-winglet—could provide a practical strategy for improving overall aerodynamic performance.

## 5. Conclusions

This study evaluated the aerodynamic performance of a blended multi-winglet configuration applied to a transonic transport aircraft under both transonic and subsonic conditions. The analysis focused on total drag reduction in the transonic regime and lift enhancement in the subsonic regime. As a result, the multi-winglet can enhance lift performance under takeoff and landing conditions while maintaining comparable performance to the single winglet during cruising. The principal findings are summarized below:Transonic conditions:Relative to the single-winglet configuration, the multi-winglet exhibited an increase in total drag of approximately three counts under cruise conditions. This increase was primarily attributed to greater profile drag, due to the enlarged frontal projected area, and additional wave drag induced by shock waves on the upper surfaces of the winglets. Conversely, the induced drag decreased by approximately 3.4 counts, primarily due to the shorter chord length of each winglet, which hinders the formation of wingtip vortices. Additionally, the negative cant angle of the rearmost winglet helps mitigate upward flow from the lower to the upper surface of the main wing, thereby reducing vortices at the wing–winglet junction. These results indicate that further geometric refinement of the multi-winglet may enable greater induced drag reduction, potentially offsetting the rise in other drag components. A future challenge under transonic conditions is to optimize the shape of the multi-winglet configuration to reduce profile drag or induced drag, thereby contributing to an overall reduction in total drag.Subsonic conditions:The multi-winglet consistently generated higher lift than the single winglet across most angles of attack. Prior to stall, this enhancement was primarily driven by the smaller leading-edge curvature of each winglet, which intensified flow acceleration over the upper surfaces. In the post-stall regime, the increased lift was attributed to leading-edge vortices that helped reattach separated flow to the upper surface, driven by accelerated airflow through the gaps between winglets. Reducing the spacing between the winglets enhances airflow across the entire upper surfaces, this effect is expected to be further amplified. In addition, the multi-winglet demonstrated improved wake vortex suppression compared to the single winglet, suggesting potential benefits for airport operations. However, the associated increase in bending moment at the wingtip—due to higher lift—may offset these benefits from a structural perspective, while the multi-winglet configuration offers notable aerodynamic advantages. Therefore, to compensate for its disadvantages and achieve improved lift performance, it is necessary to increase the area of the winglet relative to the main wing since lift is closely related to the projected area.

Based on the above, future studies should focus on optimizing the shape and arrangement of the multi-winglet for specific flight conditions. Furthermore, when applying to transonic aircraft, the integration of morphing technology may enable adaptability across a wide range of flight speeds, potentially improving overall aerodynamic performance. 

## Figures and Tables

**Figure 1 biomimetics-10-00522-f001:**
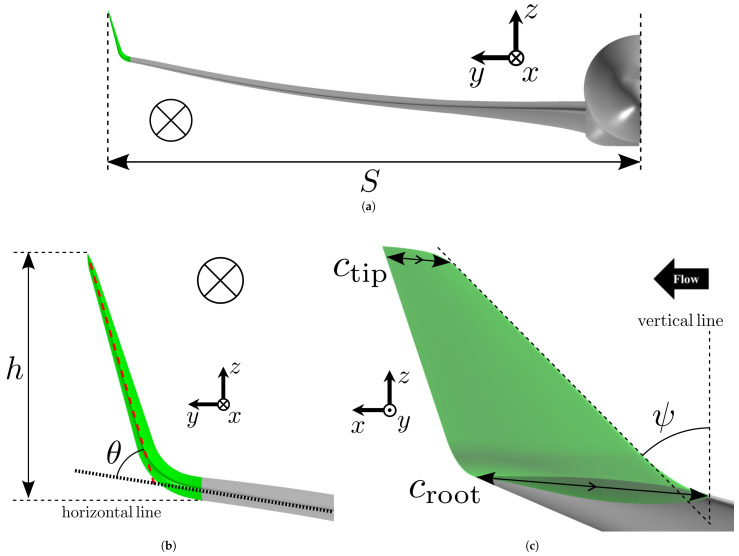
Definition of geometric parameters for the single-winglet configuration. (**a**) Front view of the aircraft. (**b**) Enlarged front view highlighting the single winglet and main wing leading edges. The red and black dotted lines indicate the leading edges of the single winglet and the main wing, respectively. (**c**) Side view corresponding to the zoomed-in region in (**b**).

**Figure 2 biomimetics-10-00522-f002:**
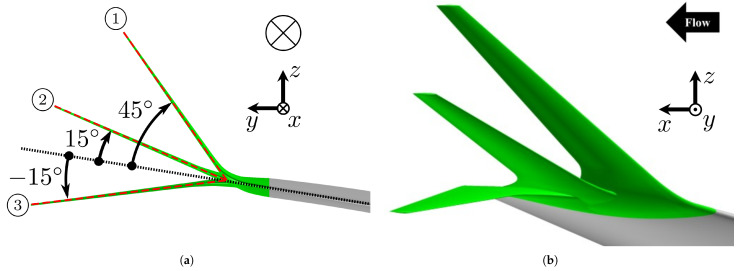
Two orthographic views of the multi-winglet configuration. (**a**) Enlarged front view highlighting the leading edges of the winglets and main wing. The red and black dotted lines indicate the leading edges of each winglet and the main wing, respectively. (**b**) Side view corresponding to (**a**).

**Figure 3 biomimetics-10-00522-f003:**
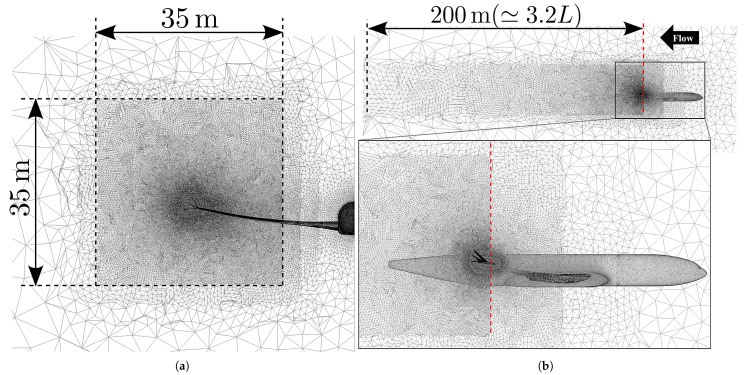
Cross-sectional views of the refined volume mesh. (**a**) Cross-section at the trailing-edge location of the main wingtip. (**b**) Visualization of the refinement region, where the red dotted line denotes the leading-edge position of the main wingtip.

**Figure 4 biomimetics-10-00522-f004:**
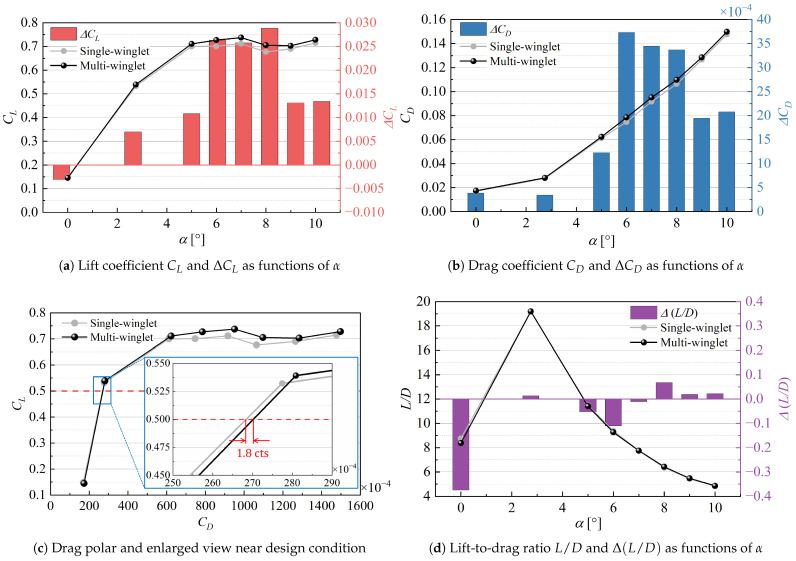
Aerodynamic coefficient comparison between the single- and multi-winglet configurations under transonic conditions.

**Figure 5 biomimetics-10-00522-f005:**
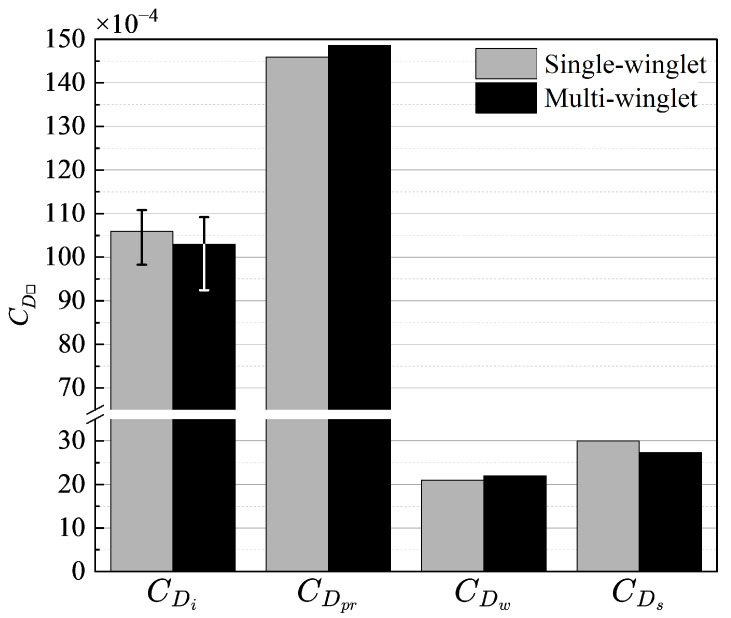
Decomposition of drag components for the single- and multi-winglet configurations under transonic cruise conditions. Error bars on Induced drag $C_{D_{i}}$ indicate the range of spatially averaged values due to downstream fluctuations beyond the refined mesh region.

**Figure 6 biomimetics-10-00522-f006:**
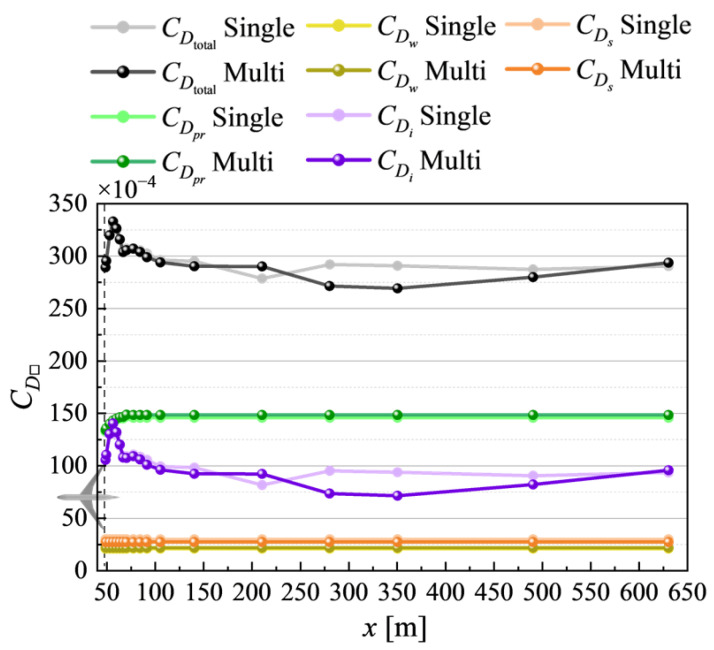
Spatial evolution of drag decomposition components in the streamwise (*x*) direction at α=2.75°. The curves show the convergence behavior of profile drag $C_{D_{pr}}$, wave drag $C_{D_{w}}$, $C_{D_{i}}$, and spurious drag $C_{D_{s}}$ as a function of wake plane location.

**Figure 8 biomimetics-10-00522-f008:**
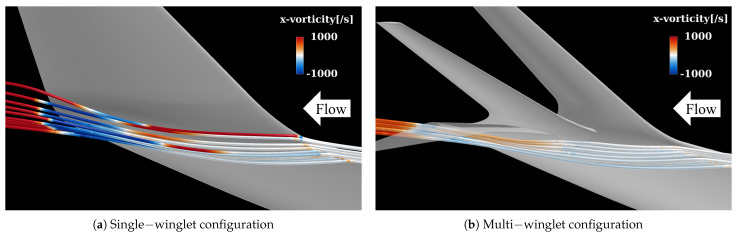
Streamlines colored by streamwise vorticity η near the junction between the main wing and the multi−winglet, visualized from the underside of the wing along the positive y− axis. In (**b**), the 3rd winglet is rendered semi−transparent to reveal flow structures beneath it.

**Figure 9 biomimetics-10-00522-f009:**
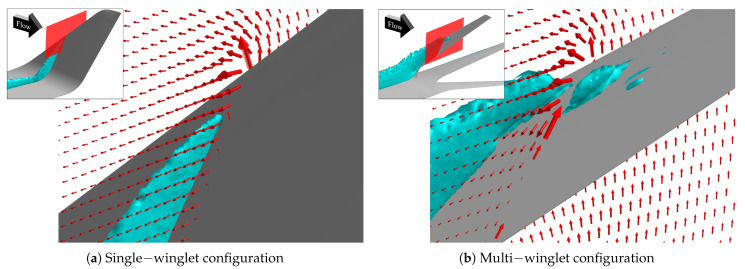
Isosurfaces of the shock detection function ($f_{\mathrm{shock}}=1.0$) and velocity vector distributions in the *y*-*z* cross-section located immediately downstream of the shock wave surface (highlighted in red in the upper−left panels).

**Figure 10 biomimetics-10-00522-f010:**
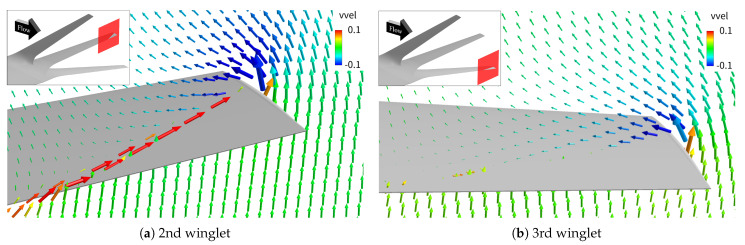
*y*–*z* cross-sectional velocity vector distributions at 50% chord length for the 2nd and 3rd winglets. The cross-section corresponds to the red surfaces highlighted in the upper-left panels.

**Figure 11 biomimetics-10-00522-f011:**
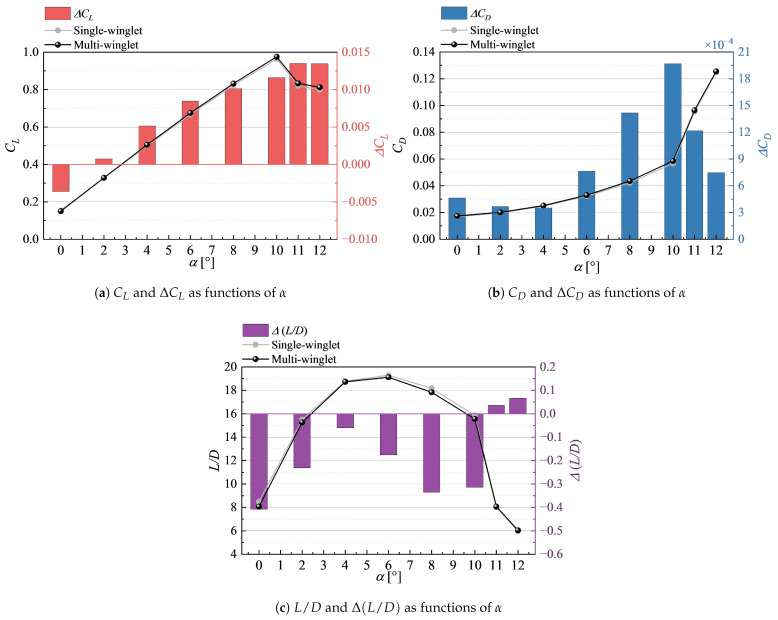
Comparison of aerodynamic coefficients for the single- and multi-winglet configurations under subsonic conditions.

**Figure 12 biomimetics-10-00522-f012:**
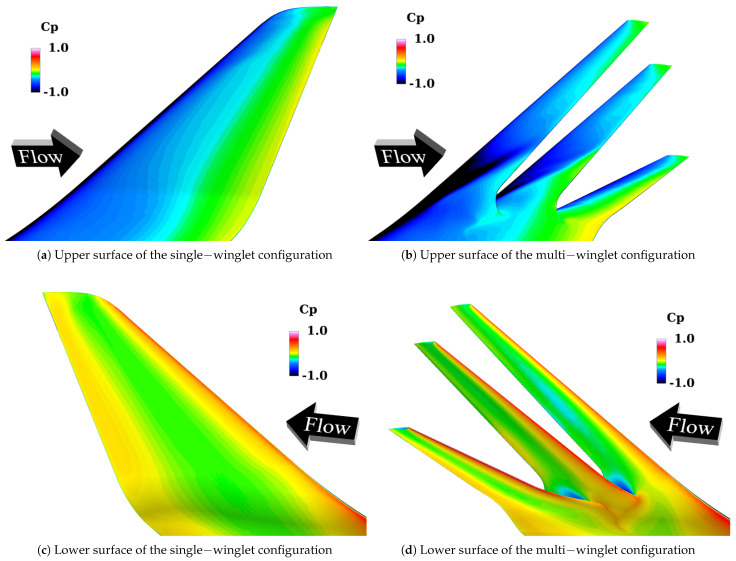
Comparison of surface pressure coefficient $C_{p}$ distributions between the single− and multi−winglet configurations at $\alpha =6^{{}^{\circ}}$.

**Figure 13 biomimetics-10-00522-f013:**
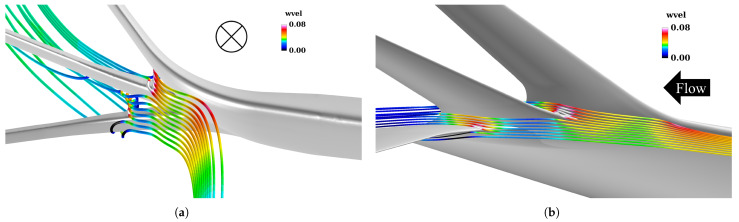
Streamline visualization near the main wingtip, colored by vertical velocity component $w_{\mathrm{vel}}$: (**a**) front view and (**b**) side view.

**Figure 14 biomimetics-10-00522-f014:**
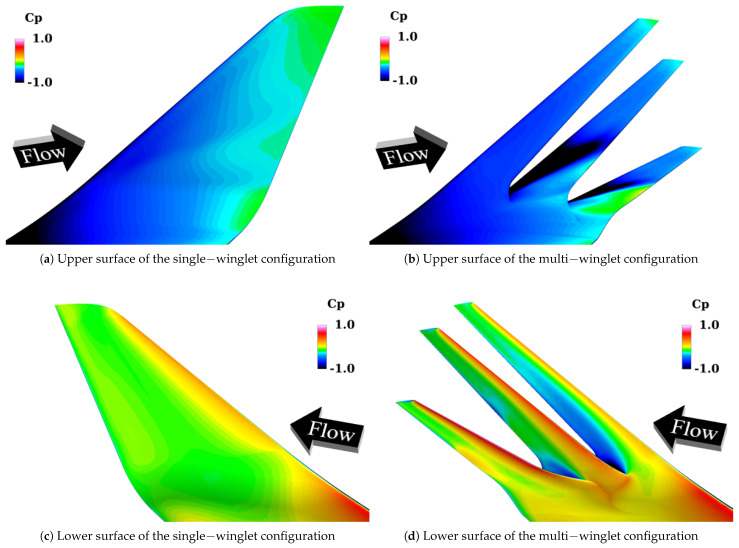
Comparison of surface $C_{p}$ distributions between the single− and multi−winglet configurations at $\alpha =12^{{}^{\circ}}$.

**Figure 15 biomimetics-10-00522-f015:**
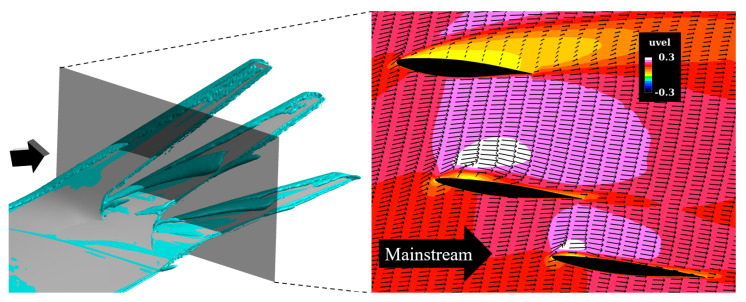
Left: Isosurfaces of Q—criterion values around the multi−winglet configuration at $\alpha =12^{{}^{\circ}}$. Right: Contours and velocity vectors of the streamwise velocity component $u_{\mathrm{vel}}$ at 25% spanwise location of the multi−winglet (black plane indicated in the left panel).

**Figure 16 biomimetics-10-00522-f016:**
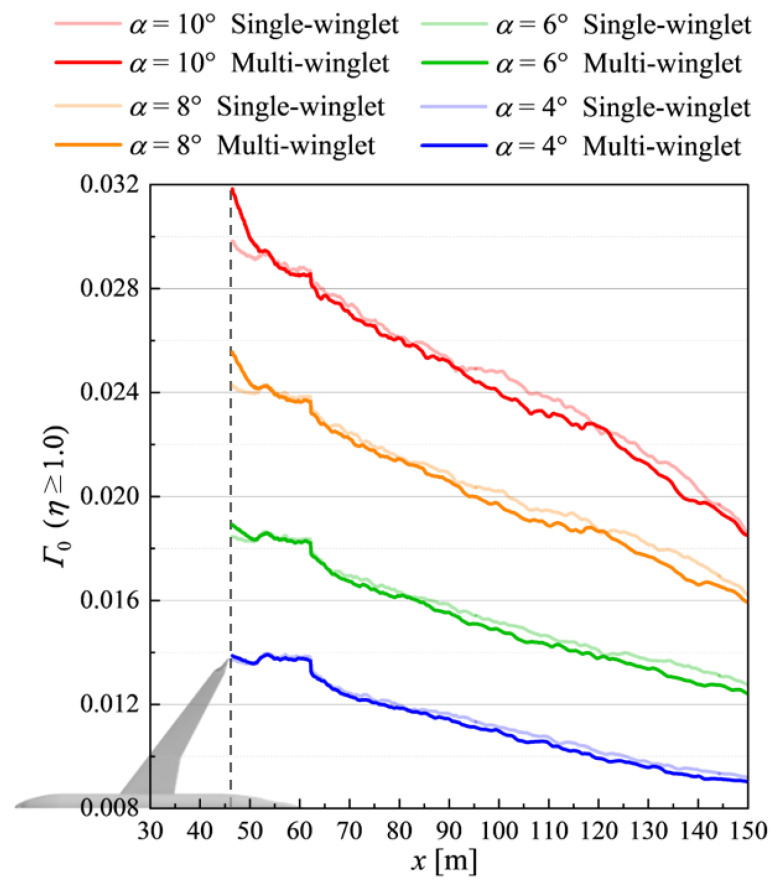
Distributions of nondimensional circulation $\Gamma_{0}$ along the *x*-direction at $\alpha =4^{{}^{\circ}}$, $6^{{}^{\circ}}$, $8^{{}^{\circ}}$, and $10^{{}^{\circ}}$. The horizontal axis represents the distance from the nose of the aircraft.

**Figure 17 biomimetics-10-00522-f017:**
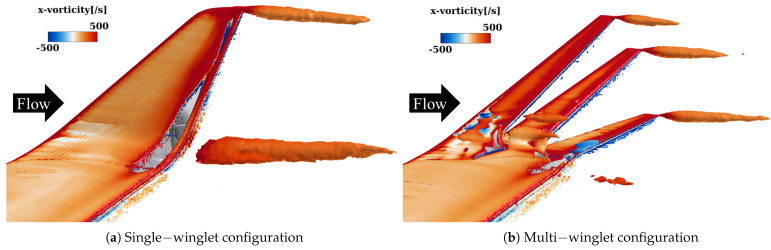
Isosurfaces of Q=0.05 visualized with vorticity η distributions at $\alpha =4^{{}^{\circ}}$ under subsonic conditions. Warm colors represent counterclockwise vortex rotation when viewed from downstream, while cool colors indicate clockwise rotation.

**Table 1 biomimetics-10-00522-t001:** Geometric parameters of the single-winglet configuration.

Parameter	Symbol	Measurement
winglet sweep angle	ψ	43°
winglet cant angle	θ	63°
winglet height	*h*	2.56 m
chord length at winglet tip	$c_{\mathrm{tip}}$	0.81 m
chord length at winglet root	$c_{\mathrm{root}}$	2.73 m

**Table 2 biomimetics-10-00522-t002:** Summary of computational conditions for subsonic and transonic regimes. The reference length used in Re is the mean aerodynamic chord.

	Subsonic	Transonic
Mach number *M*	0.2	0.85
Angle of attack α [°]	0, 2, 4, 6, 8, 10, 12	0, 2.75, 5, 6, 7, 8, 9, 10
Reynolds number Re	$5.6\times 10^{6}$	$5.0\times 10^{6}$

## Data Availability

The data that support the findings of this study are available from the corresponding author, E.K., upon reasonable request.

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
