# Peer review of "Aerodynamic Effects of a Blended Multi-Winglet on an Airliner in Subsonic and Transonic Regimes"

_biomimetics, 2025, doi:10.3390/biomimetics10080522_

Round 1
Reviewer 1 Report
Comments and Suggestions for Authors
The article is devoted to the study of the advantages of a modified wing, in which three multi-winglets are used. The studies were carried out using numerical methods. The results of the studies are very interesting. In general, the article is characterized by an unusually high quality of presentation of the results, the article is well structured, this article is extremely interesting for readers, it is recommended to publish it in the form submitted for review.
Author Response
Comments: The article is devoted to the study of the advantages of a modified wing, in which three multi-winglets are used. The studies were carried out using numerical methods. The results of the studies are very interesting. In general, the article is characterized by an unusually high quality of presentation of the results, the article is well structured, this article is extremely interesting for readers, it is recommended to publish it in the form submitted for review.
Response: We are truly grateful for your thoughtful review of our manuscript, especially given your demanding schedule.
Reviewer 2 Report
Comments and Suggestions for Authors
This study investigated the aerodynamic performance of a blended multi-winglet configuration applied to a transonic transport aircraft under both transonic and subsonic conditions based on the numerical anlysis. The studied results provide design guidelines for further CL improvement during takeoff and landing, and are applicable not only to transonic aircraft but also to subsonic configurations. Overall, this paper has theoretical and practical application value. To further improve the quality of this article, I suggest the author clarify the difficulties or differences in the study of Blended Multi Winglet in the subsonic and transonic regimes.
In this study, Mach number is 0.2 and 0.85. Why only choose these two Mach numbers? Why not choose the research range from 0.2 to 0.85?
Reviewer 3 Report
Comments and Suggestions for Authors
In this work, the authors have presented their computational studies for investigating the aerodynamic performance of a blended wing geometry with single and multiple winglets under subsonic and transonic conditions. Considering the following inputs, I believe that the manuscript needs substantial revisions before it may be recommended for publication in Biomimetics.
1. Why was the blended winglet divided into three elements? Why could it not be more or less than three?
2. Line 81: The authors claimed that they have explained mitigation in wake turbulence quantified? While understanding the use of circulation here, I could not find any details about turbulence parameters in the manuscript. Is it not more insightful to compute the turbulence parameters directly in the wake for comparisons?
3. Please provide justification for the choice of design parameters in Table 1. Are these values according to the standard reference geometry of the wing?
4. Lines 87-88: What incremental difference were mentioned here? Is it about the angle-of-attack?
5. In section 2.3, the authors should provide the governing equations for the flow dynamics. It will help readers understand the computations and the set up of steady simulations. Also, please provide details for the boundary conditions.
6. In Fig. 4a, CL starts increasing after the stall angle (8 degree). How do the authors explain it?
7. I suggest the authors to explain the drag decomposition in section 3.2, before they started discussion on their trends in this study.
8. At what section along the span of the wing did the authors plot drag in Fig. 6?
9. The authors have used two different turbulence models to compare the subsonic and transonic aerodynamic characteristics of the wings. I understand their utility for quantification of forces etc. But, explaining vortex dynamics based on RANS modeling is always tricky due to the uncertainty associated with these turbulence models. Please explain this perspective as well. For instance, what would happen to the vortex formation and its dynamics if another turbulence model is used?
Round 2
Reviewer 3 Report
Comments and Suggestions for Authors
I thank the authors for incorporaring my comments and inputs in their revised manuscript. I now recommend its publication in Biomimetics.